# A Combined Modal and Route Choice Behavioral Complementarity Equilibrium Model with Users of Vehicles and Electric Bicycles

**DOI:** 10.3390/ijerph17103704

**Published:** 2020-05-24

**Authors:** Senlai Zhu, Jie Ma, Tianpei Tang, Quan Shi

**Affiliations:** 1School of Transportation and Civil Engineering, Nantong University, Nantong 226019, China; tangtianpei@ntu.edu.cn (T.T.); shi.q@ntu.edu.cn (Q.S.); 2School of Transportation, Southeast University, Nanjing 210096, China; 3Business School, University of Shanghai for Science and Technology, Shanghai 200093, China

**Keywords:** traffic behavior, user equilibrium, complementarity problem, electric bicycle, commute mode choice

## Abstract

The popularity of electric bicycles in China makes them a common transportation mode for people to commute and move around. However, with the increase in traffic volumes for both vehicles and electric bicycles, urban traffic safety and congestion problems are rising due to traffic conflicts between these two modes. To regulate travel behavior, it is essential to analyze the mode choice and route choice behaviors of travelers. This study proposes a combined modal split and multiclass traffic user equilibrium model formulated as a complementarity problem (CP) to simultaneously characterize the mode choice behavior and route choice behavior of both vehicle and electric bicycle users. This model captures the impacts of route travel time and out-of-pocket cost on travelers’ route choice behaviors. Further, modified Bureau of Public Roads (BPR) functions are developed to model the travel times of links with and without physical separation between vehicle lanes and bicycle lanes. This study also analyzes the conditions for uniqueness of the equilibrium solution. A Newton method is developed to solve the proposed model. Numerical examples with different scales are used to validate the proposed model. The results show that electric bicycles are more favored by travelers during times of high network congestion. In addition, total system travel time can be reduced significantly through physical separation of vehicle lanes from electric bicycle lanes to minimize their mutual interference.

## 1. Introduction

In recent years, electric bicycles (E-bikes) quickly became one of the main nonmotorized travel modes in some developing countries, especially in China [1,2,3]. E-bikes have many merits over both regular bikes and vehicles, including that they are much faster and can support longer trips compared with regular bikes. Further, despite the lower maximum speed, they are much more flexible and can run at higher speeds than vehicles in congested areas. E-bikes also have other advantages, such as high availability due to low price, efficient energy consumption, and no tailpipe emissions. Due to these characteristics, E-bikes are now one of the main transportation modes in China. Each year, over 35 million E-bikes are sold in China and the installed base of E-bike in use is over 100 million [4].

However, with the increase in traffic volumes for both vehicles and electric bicycles, due to traffic conflicts between these two modes, various urban traffic problems are rising, such as traffic safety, traffic congestion, and parking problems. Recently, extensive research efforts were devoted to address the multiple research needs related to E-bikes, such as estimating cycling capacity, the bicycle equivalent unit for E-bikes, consumption behavior of E-bikes users, and characteristics of traffic accident involving E-bikes [5,6,7]. Existing research generally studied the operational characteristics of E-bike from the micro level, such as operations on links or intersections. However, as an important commute mode, few studies investigated the corresponding mode choice and route choice behaviors, which play important roles in designing management strategies and operational decisions related to E-bikes to alleviate the above traffic problems at the traffic-network level.

Similar to the Braess paradox [8], overall performance of the urban transportation network may decrease when improving the operational level of a certain link or intersection for vehicles and E-bikes. To address traffic problems caused by conflicts between vehicles and E-bikes, the mode choice and route choice behaviors of users at the network level require study, alongside the impact of various management strategies (such as separating bicycle lanes and vehicle lanes by physical separations, e.g., parterres and barriers on some links) on these travel behaviors.

Hence, to simultaneously estimate the mode choice and route choice behaviors of vehicles (cars/automobiles) and E-bikes, this paper proposes a combined modal split and route choice model, with the underlying assumption that all users own vehicles and E-bikes and can choose traffic modes based on the congestion level of each mode. Thus, the general flow pattern is estimated using an user equilibrium model which assumes that nobody can reduce their own travel cost by unilaterally shifting their route or mode at the equilibrium state. Considering the well-established complementarity theory, the proposed multimode traffic assignment model is developed as a complementarity problem (CP).

In reality, vehicle lanes and bicycle lanes on some links are separated by physical barriers, while some links have no physical separations. Link cost functions are different under these two cases. In the literature, the Bureau of Public Roads (BPR) cost function [9] was frequently adopted to analyze the link travel time characteristic of vehicles in congested networks. Without loss of generality, modified BPR functions of each mode are developed to capture the interference between vehicles and E-bikes in different scenarios. As discussed before, compared with vehicles, E-bikes can run at higher speeds in congested areas, even though their designed speeds are less than those of vehicles. This characteristic can be captured by setting different parameters in the modified BPR functions. Out-of-pocket costs are considered in the proposed model to capture impacts of factors such as fuel or electricity consumption, vehicle or E-bike depreciation, insurance, the environment, etc., on mode choice.

In summary, the contributions of this paper include:(1)The proposal of a complementarity problem to model the multimode and traffic assignment problems regarding both vehicles and E-bikes, with the uniqueness of solution to this problem analytically discussed;(2)The development of modified BPR functions to capture interactions between the two traffic modes and impacts of characteristics of each mode on link travel time;(3)The consideration of out-of-pocket costs related to factors (such as security, environment, distance, fuel and electricity prices, etc.) affecting users’ mode choice behaviors.

The rest of the paper is organized as follows. Related literature is reviewed in the next section. In Section 3, the CP model for the multimode traffic assignment problem is presented. Section 4 discusses the uniqueness of the equilibrium solution. In Section 5, a Newton method for solving the proposed model is developed. Section 6 applies the proposed method to two numerical examples. The last section concludes this study.

## 2. Literature Review

### 2.1. Traffic Assignment Model

Traffic assignment models are important tools in the design of effective management strategies and operational decisions to improve network performance, for example, they are usually leveraged to deploy infrastructures optimally to reduce traffic congestion [10,11,12,13] and to find optimal traffic signal settings to minimize total travel time [14,15]. Among various traffic assignment models, the user equilibrium problem is perhaps the most common due to its simple assumption of travelers’ behavior and the close-form formulation. The user equilibrium problem assumes that at the equilibrium state, nobody can reduce their own travel cost by unilaterally shifting their travel route.

Given Origin–Destination (OD) flows, the user equilibrium problem is treated traditionally as two different methods, namely, deterministic user equilibrium [16] and stochastic user equilibrium [17]. In deterministic user equilibrium methods, all users are assumed to have perfect information about route costs and all choose the routes of minimum cost. In stochastic user equilibrium methods, the traffic costs are considered as random variables and route choices are different among different users. In this paper, the proposed CP model estimates the deterministic user equilibrium condition of traffic flow for both vehicles and E-bikes.

### 2.2. Traffic Assignment Model with Multiple Modes

In reality, various modes can be chosen by travelers, such as vehicle, public transportation (bus, metro, taxi), bike (E-bike, regular bike), and so on. To estimate the use ratio of each mode, various methods are proposed, such as discrete choice models [18,19]. After giving the ratio of each, to address the traffic assignment problem with multiple modes, many studies attempted to convert the traffic flows of all other modes into standard passenger cars, followed by conversion of total traffic volume to be used as the input of demand for the traffic assignment model. However, each mode has particular characteristics (such as speed, travel time, capacity, security), which may be ignored if all modes are converted to the standard passenger cars.

In the literature, several multimode and/or multiclass traffic assignment models [20,21] and combined modal split and traffic assignment models [22,23] were proposed to model heterogeneous travel behavior. To capture the characteristics of travel cost realistically, these studies used different link cost functions for different traffic modes. However, in this paper, due to interactions between E-bikes and vehicles, travel cost functions may be asymmetric, thereby increasing the difficulty to formulate a mathematical programming problem. In the literature, traffic assignment problems with asymmetry travel cost functions were usually modeled as variational inequality (VI), complementarity, fixed-points, and entropy maximization problems [24,25,26,27,28,29,30]. Despite all these efforts, the best approach to this multimode traffic assignment problem is still under debate.

In existing multimode traffic assignment models, the demand of each mode is normally determined (or assumed to be elastic). That is, these models do not consider mode choice behaviors of users. To address this problem, we first need to study factors affecting mode choice behaviors. This paper focuses on studying the travel behaviors considering users of vehicles and E-bikes. In fact, except for travel time, users choose to use vehicles or E-bikes based on factors such as security, weather, distance, fuel and electricity prices, and so on. Similar to Liu and Li [31], we define costs related to these factors as out-of-pocket costs.

In addition, modes such as buses and taxis use the same lanes as vehicles, and modes such as metros do not interfere with vehicles in space at all. Different from these modes, E-bikes are designed to run in bicycle lanes separated from vehicle lanes. On some links, bicycle lanes and vehicle lanes are physically separated, such as by parterres and barriers. However, bicycle lanes and vehicle lanes are separated only by traffic markings instead of physically on some links. In this case, vehicles and E-bikes may occupy each other’s lanes. This interference should be considered in the traffic equilibrium problem, and the existing link travel time function cannot capture this interference.

To address the aforementioned gap in the existing research, this paper proposes a combined modal split and route choice model, in which out-of-pocket costs are considered in the mode route cost function and modified BPR functions are adopted to analyze the link travel time of vehicles and E-bikes for links with and without physical separation between vehicle lanes and bicycle lanes.

## 3. Model Formulation

Consider a strongly connected network [N, A], where N is the set of nodes and A is the set of directed links. Notations are first given as the following:

$T_{a}$, ${\bar{T}}_{a}$:Travel time on link a∈A for vehicles and E-bikes, respectively; $T_{a}^{0}$, ${\bar{T}}_{a}^{0}$ are the corresponding free-flow travel times;$v_{a}$, ${\bar{v}}_{a}$:Flow of vehicles and E-bikes on link a, respectively;$C_{a}$, ${\bar{C}}_{a}$:Capacity of link a for vehicles and E-bikes, respectively;$P_{i}^{w}$, ${\bar{P}}_{i}^{w}$:Travel time on route i between OD pair w for vehicles and E-bikes, respectively; $i\in P^{w}$, $P^{w}$ is the set of routes connecting OD pair w;
$f_{i}^{w}$, ${\bar{f}}_{i}^{w}$:Flow of vehicles and E-bikes on route i between OD pair w, respectively;f:$f={\left[\ldots ,f^{w},\ldots \right]}^{T}$ with dimension $n_{1}$ equals the total number of routes in the network, ${\left(\cdot \right)}^{T}$ denotes the transpose of either a vector or a matrix, and $f^{w}=\left[\ldots ,f_{i}^{w},\ldots \right]$ is the vehicle route flow set of OD pair w;
$\bar{f}$:$\bar{f}={\left[\ldots ,{\bar{f}}^{w},\ldots \right]}^{T}$ with dimension $n_{1}$, in which ${\bar{f}}^{w}=\left[\ldots ,{\bar{f}}_{i}^{w},\ldots \right]$ is the E-bike route flow set of OD pair w;$q^{w}$:Demand between OD pair w, including demand of vehicles and E-bikes. $q^{w}>0$
q:$q={\left[\ldots ,q^{w},\ldots \right]}^{T}$ with dimension $n_{2}$ equals the total number of OD pairs in the network;$\delta_{ia}^{w}$:Link–route incidence indicator, which is 1 if link a belongs to route i between OD pair w*,* and 0 otherwise;$\sigma_{i}^{w}$:Route–OD pair incidence indicator, which is 1 if route i belongs to OD pair w*,* and 0 otherwise; σ:The route–OD pair incidence matrix with the dimensions $n_{1}\times n_{2};$
$LL_{a}$:Length of link a;$LR_{i}$:Length of route i; $LR_{i}=\underset{a}{\sum }\delta_{ia}^{w}LL_{a};$
$\pi^{w}$:Minimum (equilibrium) travel cost between OD pair w;π:$\pi =\left[\ldots ,\pi^{w},\ldots \right]$ with dimension $n_{2}$;$l_{i}^{w}$, ${\bar{l}}_{i}^{w}$:Out-of-pocket costs (costs related to factors such as weather, distance, fuel and electricity prices) of route i between OD pair w for users of vehicles and E-bikes, respectively;$C_{i}^{w}$, ${\bar{C}}_{i}^{w}$Costs (including travel time and out-of-pocket costs) of route i between OD pair w for users of vehicles and E-bikes, respectively;C:$C={\left[\ldots ,C^{w},\ldots \right]}^{T}$ with dimension $n_{1}$, in which $C^{w}=\left[\ldots ,C_{i}^{w},\ldots \right]$ is the vehicle route cost set of OD pair w;$\bar{C}$:$\bar{C}={\left[\ldots ,{\bar{C}}^{w},\ldots \right]}^{T}$ with dimension $n_{1}$, in which ${\bar{C}}^{w}=\left[\ldots ,C_{i}^{w},\ldots \right]$ is the E-bike route cost set of OD pair w.

Suppose route cost functions $C_{i}^{w}$, ${\bar{C}}_{i}^{w}$ of route i between OD pair w for users of both vehicles and E-bikes are given, respectively. The multiclass traffic equilibrium problem considering users of vehicles and E-bikes is formulated as the following mixed complementary problem (MCP):(1)$$0\leq f_{i}^{w}⊥C_{i}^{w}-\pi^{w}\geq 0,\forall i$$
(2)$$0\leq {\bar{f}}_{i}^{w}⊥{\bar{C}}_{i}^{w}-\pi^{w}\geq 0,\forall i$$
(3)$$\underset{i\in P^{w}}{\sum }\left(f_{i}^{w}+{\bar{f}}_{i}^{w}\right)=q^{w},\forall w$$

Equations (1) and (2) are the complementary slackness conditions. That is, for each OD pair w, if the flow on route i satisfies $f_{i}^{w}\geq 0$ (${\bar{f}}_{i}^{w}\geq 0$), the route cost $C_{i}^{w}$ (${\bar{C}}_{i}^{w}$) on route i is equal to the minimal route cost $\pi^{w}$, i.e., $C_{i}^{w}=\pi^{w}$ (${\bar{C}}_{i}^{w}=\pi^{w}$). These complementary slackness conditions are consistent with the Wardrop’s user equilibrium (UE) principle, i.e., all used routes have equal and minimum travel costs, and all unused routes have equal or higher travel costs. Equation (3) is the flow conservation constraint.

To reduce the MCP formulated by Equations (1)–(3) to a pure CP, Equation (3) is reformulated as follows (the proof of the equivalent of Equations (3) and (4) is given in Appendix A):(4)$$0\leq \pi^{w}⊥\sum_{i\in P^{w}}\left(f_{i}^{w}+{\bar{f}}_{i}^{w}\right)-q^{w}\geq 0,\forall w.$$

Then the CP can be formulated in vector form as follows:(5)$$x=\left(\begin{array}{c} f\\ \bar{f}\\ \pi \end{array}\right)\geq 0,F\left(x\right)=\left(\begin{array}{c} C-\sigma \pi \\ \bar{C}-\sigma \pi \\ \sigma^{T}\left(f+\bar{f}\right)-q \end{array}\right)\geq 0,x^{T}F\left(x\right)=0.$$

Note that if both $C_{i}^{w}$ and ${\bar{C}}_{i}^{w}$ are linear functions with respect to traffic flow, CP (5) becomes a linear complementarity problem (LCP), which can be solved using algorithms such as Lemke’s method, the projected successive over relaxation iteration method, the general fixed-point iteration method, or the modulus-based matrix splitting iteration method [32,33,34,35,36]. If either $C_{i}^{w}$ or ${\bar{C}}_{i}^{w}$ are nonlinear, CP (5) becomes a nonlinear complementarity problem (NCP), which can be solved by algorithms such as Newton methods [37,38,39]. In this paper, route cost functions are assumed to be “convex combinations” of route travel time and out-of-pocket costs, and thus the proposed CP (5) becomes an NCP.

Specifically, route cost functions $C_{i}^{w}$, ${\bar{C}}_{i}^{w}$ are defined as
(6)$$C_{i}^{w}=P_{i}^{w}+l_{i}^{w},{\bar{C}}_{i}^{w}={\bar{P}}_{i}^{w}+{\bar{l}}_{i}^{w},$$
where $P_{i}^{w}$, ${\bar{P}}_{i}^{w}$ denote route travel time for vehicles and E-bikes, respectively, and $l_{i}^{w}$, ${\bar{l}}_{i}^{w}$ denote the corresponding out-of-pocket costs.

The route travel time is formulated as
(7)$$P_{i}^{w}=\sum_{a}\delta_{ia}^{w}T_{a},{\bar{P}}_{i}^{w}=\sum_{a}\delta_{ia}^{w}{\bar{T}}_{a}.$$

Out-of-pocket cost relate to fuel or electricity consumption, vehicle or E-bike depreciation, insurance, security, the environment, and so on. Similar to Liu and Li [31], we assume that the out-of-pocket cost is defined as a function of both travel time and travel distance, as follows:(8)$$l_{i}^{w}=\lambda_{m}P_{i}^{w}+\varphi_{m}LR_{i},{\bar{l}}_{i}^{w}=\lambda_{e}{\bar{P}}_{i}^{w}+\varphi_{e}LR_{i},$$
where $\lambda_{m}$ and $\lambda_{e}$ denote the monetary cost per unit time for vehicles and E-bikes, respectively, and $\varphi_{m}$ and $\varphi_{e}$ denote the monetary cost per unit distance traveled by vehicles and E-bikes, respectively. Note that values for $\lambda_{m}$, $\varphi_{m}$, $\lambda_{e},$ and $\varphi_{e}$ depend on factors such as fuel price, weather, and so on. In reality, these parameters can be calibrated by data collected by surveys of stated preference.

According to Equations (6)–(8), route costs $C_{i}^{w}$, ${\bar{C}}_{i}^{w}$ are obtained by
(9)$$C_{i}^{w}=\left(1+\lambda_{m}\right)\sum_{a}\delta_{a}^{i}T_{a}+\varphi_{m}LR_{i},{\bar{C}}_{i}^{w}=\left(1+\lambda_{e}\right)\sum_{a}\delta_{a}^{i}{\bar{T}}_{a}+\varphi_{e}LR_{i}.$$

Note that the linear or nonlinear characteristics of route cost function depend on the function of link travel time ($T_{a}$ and ${\bar{T}}_{a}$). On some urban links, bicycle lanes are separated from vehicle lanes by physical separations, so the flows of vehicles and E-bikes involve almost no interactions. The following modified BPR functions are used to model the link travel time for links with physical separations between bicycle lanes and vehicle lanes:(10)$$T_{a}=T_{a}^{0}\left(1+\alpha_{m}{\left(\frac{v_{a}}{C_{a}}\right)}^{\beta_{m}}\right)$$
(11)$${\bar{T}}_{a}={\bar{T}}_{a}^{0}\left(1+\alpha_{e}{\left(\frac{{\bar{v}}_{a}}{{\bar{C}}_{a}}\right)}^{\beta_{e}}\right)$$
where $v_{a}=\sum_{w}\sum_{i}\delta_{ia}^{w}f_{i}^{w}$ and ${\bar{v}}_{a}=\sum_{w}\sum_{i}\delta_{ia}^{w}{\bar{f}}_{i}^{w}$ represent vehicle flow and E-bike flow on link a, respectively, and $\alpha_{m}$($\alpha_{e}$) and $\beta_{m}$($\beta_{e}$) are constants defining how cost increases with traffic flow for vehicles (E-bikes). Note that, in reality, E-bikes have higher flexibility than vehicles and, despite lower maximum speed, E-bikes can maintain higher speeds than vehicles in congested areas. Hence, compared to the travel time of vehicles, the travel time of E-bikes is normally not significantly affected by traffic flow. That is, values of $\alpha_{e}$ and $\beta_{e}$ can be smaller than those of $\alpha_{m}$ and $\beta_{m}$, respectively. Specifically, in reality, the vehicle link capacity can usually be calibrated due to designed travel speed, number of lanes, headway, etc., and the E-bike link capacity can be calibrated by real link travel data [5].

In some urban links, no independent bicycle lanes or bicycle lanes are separated from vehicle lanes by traffic markings, meaning vehicle lanes may be occupied by E-bikes and bicycle lanes may also be occupied by vehicles frequently. In this case, the link travel time functions related to links without physical separations between bicycle lanes and vehicle lanes are formulated as follows:(12)$$T_{a}=T_{a}^{0}\left(1+\alpha_{m}{\left(\frac{v_{a}+\gamma_{m}{\bar{v}}_{a}}{\theta_{m}C_{a}}\right)}^{\beta_{m}}\right)$$
(13)$${\bar{T}}_{a}={\bar{T}}_{a}^{0}\left(1+\alpha_{e}{\left(\frac{{\bar{v}}_{a}+\gamma_{e}v_{a}}{\theta_{e}{\bar{C}}_{a}}\right)}^{\beta_{e}}\right)$$
where $\gamma_{m}$($\gamma_{e}$) is a constant characterizing the impact of traffic flow of E-bikes (vehicles) on travel cost of vehicles (E-bikes) and $\theta_{m}$($\theta_{e}$) is a constant defining how the link capacity of vehicles (E-bikes) is influenced by the traffic flow of E-bikes (vehicles).

Despite no physical separation, vehicles and E-bikes are designed to run in separated lanes or spaces, and $C_{a}$ and ${\bar{C}}_{a}$ represent the capacity of these designed lanes or spaces. However, due to no physical separation, one mode may occupy the designed lanes or spaces of other mode. Considering this phenomenon of occupation, parameters $\theta_{m}$ and $\theta_{e}$ are introduced to adjust the corresponding capacity, and $\gamma_{m}$ and $\gamma_{e}$ are considered to adjust the corresponding traffic flow. Since the volume of a vehicle is normally larger than that of a E-bike, $\gamma_{m}$ is assumed to be bigger than $\gamma_{e}$. In fact, link time Functions (10) and (11) regarding cases of physical separation can be treated as special forms of link time Functions (12) and (13) for cases without physical separation, respectively. In this special form, $\gamma_{m}=\gamma_{e}=0$ and $\theta_{m}=\theta_{e}=1$.

## 4. Uniqueness of Equilibrium Solution

Note that link travel time Functions (10)–(13) are nonlinear and the proposed traffic assignment model is an NCP. According to the theory of VI/CP [40], since the feasible region of our problem is convex and route cost Function (9) is continuous, if the Jacobian matrix J of route cost Function (9) is definitely positive, the solution of the proposed NCP is unique.

The corresponding Jacobian matrix J is:(14)$$J=\left[\begin{array}{ll} \frac{\partial C_{i}^{w}}{\partial f_{i}^{w}} & \frac{\partial C_{i}^{w}}{\partial {\bar{f}}_{i}^{w}}\\ \frac{\partial {\bar{C}}_{i}^{w}}{\partial f_{i}^{w}} & \frac{\partial {\bar{C}}_{i}^{w}}{\partial {\bar{f}}_{i}^{w}} \end{array}\right]=\left[\begin{array}{ll} (1+\lambda_{m})\sum_{a}\delta_{a}^{i}\dot{T_{a}} & (1+\lambda_{m})\sum_{a}\delta_{a}^{i}\ddot{T_{a}}\\ (1+\lambda_{e})\sum_{a}\delta_{a}^{i}\dot{{\bar{T}}_{a}} & (1+\lambda_{e})\sum_{a}\delta_{a}^{i}\ddot{{\bar{T}}_{a}} \end{array}\right]$$
where $\dot{T_{a}}=\frac{\partial T_{a}}{\partial v_{a}}$, $\ddot{T_{a}}=\frac{\partial T_{a}}{\partial {\bar{v}}_{a}}$, $\dot{{\bar{T}}_{a}}=\frac{\partial {\bar{T}}_{a}}{\partial v_{a}},$ and $\ddot{{\bar{T}}_{a}}=\frac{\partial {\bar{T}}_{a}}{\partial {\bar{v}}_{a}}$.

**Theorem** **1.**
*According to Equation (10) and Equation (11), the solution to the proposed NCP is unique in cases of physical separation.*


**Proof.** For cases of physical separation between vehicle lanes and bicycle lanes, $\ddot{T_{a}}=\dot{{\bar{T}}_{a}}=0$. Then, J becomes:(15)$$J=\left[\begin{array}{cc} \left(1+\lambda_{m}\right)\sum_{a}\delta_{a}^{i}\dot{T_{a}} & 0\\ 0 & \left(1+\lambda_{e}\right)\sum_{a}\delta_{a}^{i}\ddot{{\bar{T}}_{a}} \end{array}\right]$$
which is a standard diagonal matrix with positive diagonal elements. Obviously, in this case J is definitely positive. The uniqueness of the solution to the NCP under cases of physical separation is therefore proven. □ 

Next, we discuss the case of no physical separation between vehicle lanes and bicycle lanes on links of the transportation network. 

**Theorem** **2.**
*Given link time Functions (12) and (13), if*
$1-\gamma_{m}\gamma_{e}$
*> 0, the solution to the proposed NCP is unique for cases without physical separation.*


**Proof.** For cases of no physical separation between vehicle lanes and bicycle lanes, according to Equation (14), the first-order leading principal minor of J is $\left(1+\lambda_{m}\right)\sum_{a}\delta_{a}^{i}\dot{T_{a}}$, which is positive. Then we discuss the second-order leading principal minor, i.e., $D\prime_{2}$:$$D^{\prime }{}_{2}=\left|\begin{array}{cc} \left(1+\lambda_{m}\right)\sum_{a}\delta_{a}^{i}\dot{T_{a}} & \left(1+\lambda_{m}\right)\sum_{a}\delta_{a}^{i}\ddot{T_{a}}\\ \left(1+\lambda_{e}\right)\sum_{a}\delta_{a}^{i}\dot{{\bar{T}}_{a}} & \left(1+\lambda_{e}\right)\sum_{a}\delta_{a}^{i}\ddot{{\bar{T}}_{a}} \end{array}\right|\;=\left(1+\lambda_{m}\right)\left(1+\lambda_{e}\right)\left(\underset{a}{\sum }\delta_{a}^{i}\dot{T_{a}}\underset{a}{\sum }\delta_{a}^{i}\ddot{{\bar{T}}_{a}}-\underset{a}{\sum }\delta_{a}^{i}\ddot{T_{a}}\underset{a}{\sum }\delta_{a}^{i}\dot{{\bar{T}}_{a}}\right)$$Thus, $D\prime_{2}>0$ if, and only if,
(16)$$\underset{a}{\sum }\delta_{a}^{i}\dot{T_{a}}\underset{a}{\sum }\delta_{a}^{i}\ddot{{\bar{T}}_{a}}-\underset{a}{\sum }\delta_{a}^{i}\ddot{T_{a}}\underset{a}{\sum }\delta_{a}^{i}\dot{{\bar{T}}_{a}}>0$$In Equation (16), $\dot{T_{a}}=\frac{\alpha_{m}\beta_{m}T_{a}^{0}}{\theta_{m}C_{a}}{\left(\frac{v_{a}+\gamma_{m}{\bar{v}}_{a}}{\theta_{m}C_{a}}\right)}^{\beta_{m}-1}$, $\ddot{{\bar{T}}_{a}}=\frac{\alpha_{e}\beta_{e}{\bar{T}}_{a}^{0}}{\theta_{e}C_{a}}{\left(\frac{{\bar{v}}_{a}+\gamma_{e}v_{a}}{\theta_{e}{\bar{C}}_{a}}\right)}^{\beta_{e}-1}$, $\ddot{T_{a}}=\frac{\alpha_{m}\beta_{m}\gamma_{m}T_{a}^{0}}{\theta_{m}C_{a}}{\left(\frac{v_{a}+\gamma_{m}{\bar{v}}_{a}}{\theta_{m}C_{a}}\right)}^{\beta_{m}-1}$, and $\dot{{\bar{T}}_{a}}=\frac{\alpha_{e}\beta_{e}\gamma_{e}{\bar{T}}_{a}^{0}}{\theta_{e}C_{a}}{\left(\frac{{\bar{v}}_{a}+\gamma_{e}v_{a}}{\theta_{e}{\bar{C}}_{a}}\right)}^{\beta_{e}-1}$. Denote $A^{1}=\frac{\alpha_{m}\beta_{m}T_{a}^{0}}{\theta_{m}C_{a}}{\left(\frac{v_{a}+\gamma_{m}{\bar{v}}_{a}}{\theta_{m}C_{a}}\right)}^{\beta_{m}-1}$ and $A^{2}=\frac{\alpha_{e}\beta_{e}{\bar{T}}_{a}^{0}}{\theta_{e}C_{a}}{\left(\frac{{\bar{v}}_{a}+\gamma_{e}v_{a}}{\theta_{e}{\bar{C}}_{a}}\right)}^{\beta_{e}-1}$, then:(17)$$\underset{a}{\sum }\delta_{a}^{i}\dot{T_{a}}\underset{a}{\sum }\delta_{a}^{i}\ddot{{\bar{T}}_{a}}-\underset{a}{\sum }\delta_{a}^{i}\ddot{T_{a}}\underset{a}{\sum }\delta_{a}^{i}\dot{{\bar{T}}_{a}}=\left(1-\gamma_{m}\gamma_{e}\right)A^{1}A^{2}$$Thus, if $1-\gamma_{m}\gamma_{e}$ > 0, $D\prime_{2}>0$. □

In summary, if $1-\gamma_{m}\gamma_{e}$ > 0, all order leading principal minors of J are positive, then the solution to the proposed NCP is unique for cases of no physical separation. 

Note that values of $\gamma_{m}$ and $\gamma_{e}$ can impact the corresponding traffic flows of each mode due to interactions between flows of vehicles and E-bikes. In the most extreme case, vehicle lanes are occupied by all vehicles and E-bikes, and bicycle lanes are also occupied by all vehicles and E-bikes. In this case, $\gamma_{m}$ can be treated as the vehicle equivalent of an E-bike (denoted as VE) and $\gamma_{e}$ can be treated as the E-bike equivalent of a vehicle (denoted as EE). According to the conversion relationship, VE × EE = 1. In reality, despite no physical separation between vehicle lanes and bicycle lanes, some vehicles and E-bikes still run in the corresponding designed lanes, with not all vehicles (E-bikes) occupying the bicycle (vehicle) lane. Thus, $\gamma_{m}<\mathrm{VE}$ and $\gamma_{e}<\mathrm{EE}$. Therefore, $1-\gamma_{m}\gamma_{e}>1-\mathrm{VE}*\mathrm{EE}=0$, that is, the condition $1-\gamma_{m}\gamma_{e}$ > 0 in Theorem 2 is satisfied. In summary, the unique solution to the proposed NCP can be guaranteed in reality.

## 5. Solution Algorithm

Several algorithms were proposed in the literature to solve the traffic assignment problem formulated as an NCP. Among these algorithms, the nonsmooth and semismooth Newton methods were widely used, the basic idea being to convert the complementarity problem into an equal system of equations so as to solve them using the general Newton method.

First, we give the following definition.

**Definition** **1.**
*A function $\phi :R^{2}\to R$ is called an NCP function if*
(18)ϕ(a,b)⇔ab=0,a≥0,b≥0


According to Definition 1, the NCP function related to NCP (5) can be defined as
(19)$$\phi \left(x\right)=\left(\begin{array}{c} \phi \left(x_{1},F_{1}\left(x\right)\right)\\ \vdots \\ \phi \left(x_{n},F_{n}\left(x\right)\right) \end{array}\right)$$

Then, the solution to NCP (5) can be obtained by solving ϕ(x)=0.

Note that the NCP function significantly impacts the effective solution algorithm. The following Fischer–Burmeister (FB) function [38] is frequently used as the NCP function.
(20)$$\phi_{FB}\left(a,b\right)=a+b-\sqrt{a^{2}+b^{2}}$$

The FB Function (20) has many interesting properties, however, it is too flat in the positive orthant (the main region of interest for a complementarity problem) when dealing with a monotone complementarity problem. Chen et al. [37] introduced another NCP function, i.e.,
(21)$$\phi_{\lambda }\left(a,b\right)=\lambda \phi_{FB}\left(a,b\right)+\left(1-\lambda \right)a_{+}b_{+}$$
where λ∈(0,1] is an arbitrary parameter and $a_{+}b_{+}$ are penalties for violating the complementarity conditions, in which, for example, $z_{+}$ is a nonnegative operator $z_{+}=max\left(0,z\right)$ for ∀z∈R.

Based on the NCP Function (21), the equal system of equation related to NCP Function (5) is defined as
(22)$$\phi_{\lambda }\left(x\right)=\left(\begin{array}{c} \phi_{\lambda }\left(x_{1},F_{1}\left(x\right)\right)\\ \vdots \\ \phi_{\lambda }\left(x_{n},F_{n}\left(x\right)\right) \end{array}\right)$$

Further, we define
(23)$$\varphi_{\lambda }\left(a,b\right)=\frac{1}{2}\phi_{\lambda }{\left(a,b\right)}^{2}$$

Then, a natural merit function $\psi_{\lambda }\left(x\right)$ of $\phi_{\lambda }\left(x\right)$ is given by
(24)$$\psi_{\lambda }\left(x\right)=\frac{1}{2}\|\phi_{\lambda }\left(x\right){\|}^{2}=\overset{n}{\underset{i=1}{\sum }}\varphi_{\lambda }\left(x_{i},F_{i}\left(x\right)\right)$$
where ‖·‖ is the Euclidean norm.

The NCP Function (21) and the merit Function (24) were proven by Chen et al. [37] and Xu et al. [39] to possess all the positive features of the FB Function (20) and its corresponding merit function.

Thus, the Newton method introduced by Du Luca et al. [41] to solve the proposed NCP Function (5) with NCP Function (21) and merit Function (24) is given as follows.

**Algorithm 1**. Global algorithm.**Step 1.1**. Initialize parameters μ∈(0,1), $\omega \in \left(0,\frac{1}{2}\right)$, ρ>0, p>2, tolerance error ε>0 to check convergence, iteration counter k=0.
**Step 1.2.** Initialize solution vector $x^{0}=\left(\begin{array}{c} f^{0}\\ {\bar{f}}^{0}\\ \pi^{0} \end{array}\right)$.
**Step 1.3.** If $\|\nabla \psi_{\lambda }\left(x^{k}\right)\|\leq \varepsilon$, then terminate. Otherwise, go to the next step.
**Step 1.4.** Choose $V_{k}$ from the C-subdifferential $\partial_{C}\phi_{\lambda }{\left(x^{k}\right)}^{T}$ of $\phi_{\lambda }\left(x^{k}\right)$ and let $d_{k}\in R^{2n_{1}+n_{2}}$ be a solution of the following linear system of equations: (25)$$V_{k}d=-\phi_{\lambda }\left(x^{k}\right).$$ If solution $d_{k}$ cannot be found or if the descent test (26)$$\nabla \psi_{\lambda }{\left(x^{k}\right)}^{T}d_{k}\leq -\rho \|d_{k}{\|}^{p}$$ does not satisfy, set $d_{k}=-\nabla \psi_{\lambda }{\left(x^{k}\right)}^{T}$.
**Step 1.5.** Linear search. Find the smallest nonnegative integer $l^{k}$ such that (27)$$\psi_{\lambda }\left(x^{k}+\mu^{l^{k}}\right)\leq \psi_{\lambda }\left(x^{k}\right)+\omega \mu^{l^{k}}\nabla \psi_{\lambda }{\left(x^{k}\right)}^{T}d_{k}$$
**Step 1.6.** Set $x^{k+1}=x^{k}+\mu^{l^{k}}d_{k}$, k=k+1 and go to Step 1.3.

Step 1.2 is for the initialization of the solution vector $x^{0}$. Although we can set the arbitrary vector as the initial solution vector, to make the global algorithm more efficient in solving the proposed NCP Function (5), the following procedure (Algorithm 2) is used to initialize the solution vector.

**Algorithm 2**. Initialize the solution vector.**Step 2.1**. Read in a predefined route set $P^{w}$. Choose any initial vehicle demand $q_{v}^{w}$ and E-bike demand $q_{e}^{w}$, $q_{v}^{w}+q_{e}^{w}=q^{w}$.
**Step 2.2**. Load demand $q_{v}^{w}$ to route set $P^{w}$ using the all-or-nothing method to obtain an initial vehicle route flow vector $f^{0}$.
**Step 2.3**. Update the link time and route cost $C_{i}^{w}$ according to Equations (9)–(13).
**Step 2.4**. Load demand $q_{e}^{w}$ to route set $P^{w}$ using the all-or-nothing method to obtain an initial E-bike route flow vector ${\bar{f}}^{0}$.
**Step 2.5**. Update the route cost $C_{i}^{w}$.
**Step 2.6**. Select the initial min-route cost $\pi^{w,0}=min\left\{C_{i}^{w},i\in R^{w}\right\}$ and set $\pi^{0}={\left[\ldots ,\pi^{w,0},\ldots \right]}^{T}$.
**Step 2.7**. Set the initial solution vector $x^{0}=\left(\begin{array}{c} f^{0}\\ {\bar{f}}^{0}\\ \pi^{0} \end{array}\right)$.

Further, in Step 1.4 of the global algorithm (Algorithm 1), we first need to choose $V_{k}$ from the C-subdifferential $\partial_{C}\phi_{\lambda }{\left(x^{k}\right)}^{T}$ (see [37,39] for definition). Similar to Chen et al. [37], we use the following Algorithm 3 to choose $V_{k}$.

**Algorithm 3**. Choose Vk∈∂Cϕλ(xk)T**Step 3.1**. Let x∈R2n1+n2 be given and Vi denote the *i*th row of a matrix V∈R(2n1+n2)×(2n1+n2).
**Step 3.2**. Set index set S1={i|xi=Fi(x)=0} and S2={i|xi>0,Fi(x)>0}.
**Step 3.3**. Set z∈R2n1+n2 such that zi=0 for i∉S1 and zi=1 for i∈S1.
**Step 3.4**. Set Vi as follows:If i∈S1, set (28)Vi=λ(1−zi‖(zi,∇Fi(x)Tz)‖)eiT+λ(1−∇Fi(x)Tz‖(zi,∇Fi(x)Tz)‖)∇Fi(x)T. If i∈S2, set (29)Vi=[λ(1−xi‖(xi,Fi(x))‖)+(1−λ)Fi(x)]eiT+[λ(1−Fi(x)‖(xi,Fi(x))‖)+(1−λ)xi]∇Fi(x)T If i∉S1∪S2, set (30)Vi=λ(1−xi‖(xi,Fi(x))‖)eiT+λ(1−Fi(x)‖(xi,Fi(x))‖)∇Fi(x)T

## 6. Numerical Examples

In this section, we present numerical examples to verify the proposed model and analyze the users’ mode (vehicle and E-bike) choice behavior.

### 6.1. A Simple Example

First, we apply the proposed NCP to a simple network as shown in Figure 1. The network has five nodes, five links, and two OD pairs (1-5 and 2-5). Link parameters are shown in Table 1. Demands of OD pairs 1-5 and 2-5 are set to be 300 and 200, respectively. Parameters in link time Functions (10)–(13) are set to be $\alpha_{m}=0.15$, $\beta_{m}=4$,$\alpha_{e}=0.1$, $\beta_{e}=2$, $\gamma_{m}=0.3$, and $\gamma_{e}=3$, for any link, and parameters in route cost Function (9) are set to be $\lambda_{m}=0.1$, $\varphi_{m}=0.2$, $\lambda_{e}=0.2$, $\varphi_{e}=0.4$, and $\theta_{m}=\theta_{e}=1.1$ for any route.

Table 2 shows the results of link flows for two cases. The first case demonstrates no physical separation between the vehicle lane and the bicycle lane for any link in the network. The second case regards all links in the network having physical separations between vehicle lanes and bicycle lanes. As can be seen from Table 2, the link flow patterns are quite different for the two cases, indicating that physical separation between the vehicle lane and the bicycle lane has dramatic effects on the equilibrium flow pattern. Table 3 shows the route flow and route cost of both vehicles and E-bikes, denoting that (a) the sum of route flows of vehicles and E-bikes for each OD pair is equal to the total demand of that OD pair, and (b) for any OD pair, the cost of each used route (including vehicles and E-bikes) is equal to its minimum travel cost (in bold) and the unused routes have higher costs than the minimum OD travel cost.

Table 4 further compares some indicators of the flow patterns for cases with and without physical separation, including demand of vehicles, demand of E-bikes, and total travel time of the system. It can be seen from Table 4 that mode choice proportions are significantly influenced by physical separation between vehicle lanes and bicycle lanes. The total system travel time for cases with physical separation is much smaller than in cases without physical separation. This is because physical separation decreases the interference between vehicles and E-bikes, thereby reducing travel costs for both traffic modes.

### 6.2. Sensitivity Analysis

Figure 2 shows how mode choice proportions vary from the total demand. In this experiment, demands of OD pairs 1-5 and 2-5 are set to grow at the same rate. It can be seen from Figure 2 that proportion of vehicle demand decreases with increasing total demand, while the proportion of E-bike demand changes conversely. If the total demand is larger, the network tends to be more congested. As discussed before, E-bikes have higher flexibility than vehicles and can maintain higher speeds than vehicles in congested areas despite their lower maximum speeds. Hence, when the network is more congested, users of the network prefer to choose E-bikes rather than vehicles.

Figure 3 shows how proportions of vehicle and E-bike demand vary by changes in values of out-of-pocket-related parameters $\varphi_{m}$, $\lambda_{m}$, $\varphi_{e}$, and $\lambda_{e}$. As shown in Equation (8), $\varphi_{m}$ and $\varphi_{e}$ denote the monetary cost per unit of distance traveled by vehicles and E-bikes, respectively, and $\lambda_{m}$ and $\lambda_{e}$ denote the monetary cost per unit of time for vehicles and E-bikes, respectively. Figure 3 shows that the proportion of vehicle (E-bikes) demand decreases with the growth of the corresponding parameters $\lambda_{m}$ and $\varphi_{m}$ ($\lambda_{e}$ and $\varphi_{e}$). As mentioned before, out-of-pocket cost is impacted by fuel or electricity consumption, vehicle or E-bike depreciation, insurance, the environment, and so on. For example, when the price of fuel increases (values of $\varphi_{m}$ and $\lambda_{m}$ increase in this case), users tend to reduce vehicle use. Similarly, when the weather conditions are poor (such as raining) or the road safety conditions for nonmotor vehicles are low (values of $\varphi_{e}$ and $\lambda_{e}$ increase in these cases), out-of-pocket costs related to E-bikes increase, thus the choice proportion of E-bikes becomes smaller.

Table 5 shows the sensitivity analysis results of parameters related to interactions between vehicles and E-bikes ($\gamma_{m}$ and $\gamma_{e}$ in Equations (12) and (13)). In reality, the influence between flows of vehicles and E-bikes is mutual, that is, when the impact on vehicles from E-bikes becomes greater, the impact on E-bikes from vehicles also becomes greater simultaneously. Hence, in Table 5, the corresponding parameters $\gamma_{m}$ and $\gamma_{e}$ are shown to increase equidistantly. The results show that the total system total travel time increases with the degree of interaction between vehicles and E-bikes. The total travel time of each mode also tends to increase with the degree of interaction. Combined with results of Table 4, flow of vehicles and E-bikes should be physically separated as much as possible, and interference between vehicles and E-bikes should also be reduced as much as possible in links without physical separation between vehicle lanes and bicycle lanes, such as by regulating traffic.

### 6.3. A Larger Network

The proposed model is further tested using the well-known Nguyen–Dupuis network shown in Figure 4. It consists of 13 nodes, 38 bidirectional links, and 18 OD pairs (all possible combinations between the three left nodes {12, 1, 4} and the three right nodes {8, 2, 3}). Fifty routes considered by Zhu et al. [42] are used. The length of each link is set to be $LL_{a}=10$, and other link characteristic parameters are shown in Table 6. Parameters in link time Functions (12) and (13) are set to be $\gamma_{m}=0.2$ and $\gamma_{e}=2$ for any link, and other parameters in link time Functions (10)–(13) and route cost Function (9) are set to be the same as those of the simple example. The travel demands of 18 OD pairs are shown in Table 7.

As discussed before, cases of physical separation are special cases of those without physical separation, with the proposed model only tested for cases of no physical separation between vehicle lanes and bicycle lanes for any link in the network. Table 8 shows the resulting link flow pattern, where, for any link, the link flow of the E-bike is larger than that of vehicle. Hence, given the OD demand shown in Table 7, the proportion of E-bike demand is larger than that of vehicle demand. The simple example proves that the proportion of vehicle demand decreases with increasing total demand, while the proportion of E-bike demand increases with the total demand. Similarly, when we reduce the total travel demand of the Nguyen–Dupuis network, vehicle choice proportion increases and E-bike choice proportion decreases.

To check the correctness of the solutions when applying the proposed NCP model to a larger network, Table 9 shows the resulting route flows and route costs. Without loss of generality, Table 9 only shows the results for OD pairs 1-3 and 4-2. First, the sum of route flows of vehicles and E-bikes for each OD pair should equal the total demand of that OD pair. In detail, for OD pair 1-3, the total vehicle route flow is 2.36 + 19.00 + 46.86 = 68.22 and the total E-bike route flow is 129.01 + 19.13 + 126.39 + 12.39 + 74.76 = 361.78. Hence, the total flow is 68.22 + 361.78 = 430, which equals the travel demand of OD pair 1-3. Thus, the sum of vehicle and E-bike route flows for each OD pair should equal to the total demand of the OD pair, which is satisfied for OD pair 1-3. Similarly, OD pair 4-2 is also verified. Second, for any OD pair, the cost of each used route (including vehicles and E-bikes) should equal its minimum travel cost, with the unused routes having higher costs than the minimum OD travel cost. Table 9 clearly shows that the resulting route flows and costs of vehicles and E-bikes satisfy the Wardrop UEprinciple.

## 7. Conclusions

To consider the mode choice behavior between vehicles and E-bikes, this paper presented a multimode traffic assignment model formulated as a CP. In this model, the route cost was assumed to consist of route travel time and out-of-pocket cost. Then, the BPR link travel time function was extended for cases with and without physical separation between vehicle lanes and bicycle lanes, respectively. Given the modified link travel time functions, the proposed multimode traffic assignment model became an NCP. Further, the uniqueness of the equilibrium solution to the proposed NCP was proven and a solution algorithm based on the Newton method was developed.

Numerical examples were conducted on two different size networks, with the results showing that the solution to the proposed NCP was correct, that is, the solution satisfied the Wardrop UE principle and flow conservation law. Sensitivity analysis showed that since E-bikes are more flexible than vehicles, especially in congested areas, E-bike choice proportion increased with total travel demand and vehicle choice proportion decreased accordingly. Increased in out-of-pocket costs led to a reduction of the corresponding mode choice.

Through the comparison of cases with and without physical separations, it was found that reducing the inference between vehicles and E-bikes could significantly reduce total system travel time, indicating that traffic engineers and planners should try to physically separate the flow of vehicles and E-bikes as much as possible.

In reality, due to factors such as resource constraints, it is impossible for all links to be physically separated between vehicle lanes and bicycle lanes. Hence, in future research, it would be worthwhile to optimize which links should set up physical separations and extend the proposed CP model to the network design problem. Meanwhile, the application of the proposed model should be tested on real networks, and the modified link travel time function should be verified and calibrated by real link travel data. In addition, some travelers are accustomed to certain traffic modes and do not consider other modes in reality, and some travelers may have more alternatives besides vehicles and E-bikes, such as public transportation. Hence, extending the proposed CP model to consider more types of travelers in the future would be worthwhile.

## Figures and Tables

**Figure 1 ijerph-17-03704-f001:**
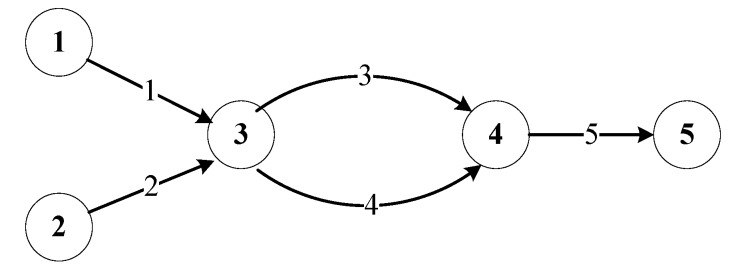
A simple network.

**Figure 2 ijerph-17-03704-f002:**
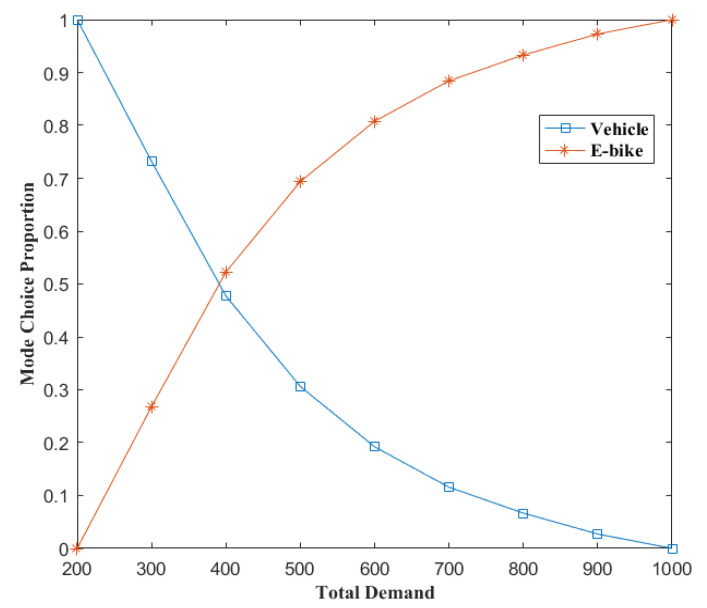
Mode choice proportions change with total demand for the example network.

**Figure 3 ijerph-17-03704-f003:**
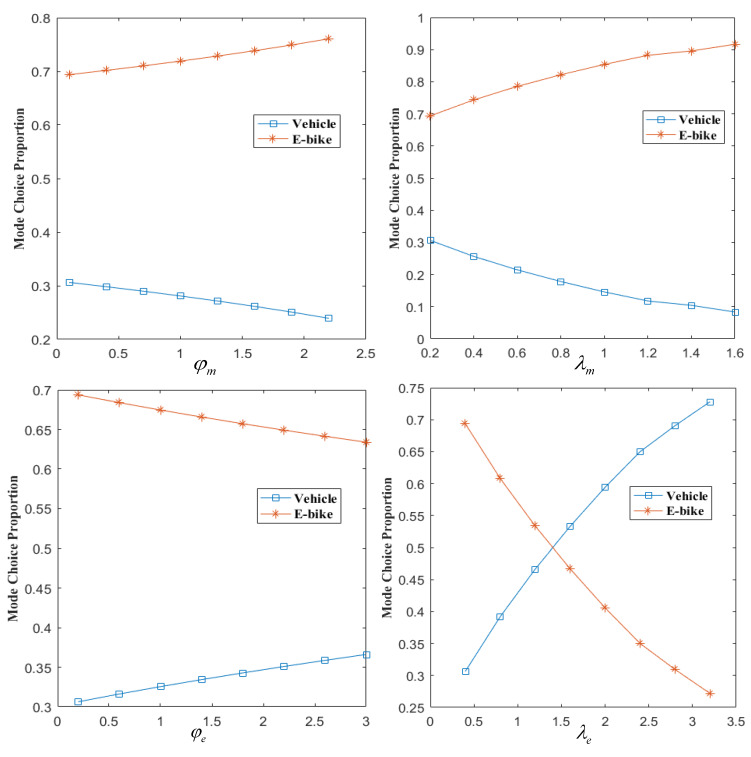
Sensitivity analysis regarding parameters related to out-of-pocket cost.

**Figure 4 ijerph-17-03704-f004:**
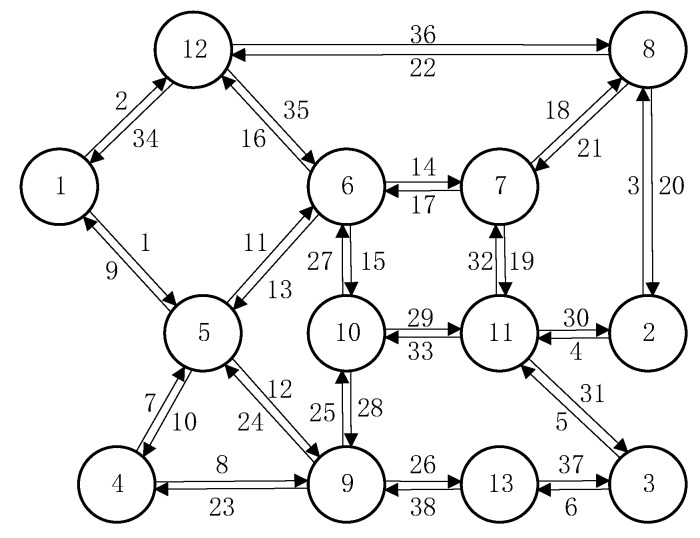
The Nguyen–Dupuis network.

**Table 1 ijerph-17-03704-t001:** Link parameters of network in Figure 1.

Link	1	2	3	4	5
$T_{a}^{0}$	10	10	20	15	10
${\bar{T}}_{a}^{0}$	20	20	40	30	20
$C_{a}$	40	50	30	60	80
${\bar{C}}_{a}$	60	80	50	100	120
$LL_{a}$	10	10	15	10	10

**Table 2 ijerph-17-03704-t002:** Link flows of the nonlinear complementarity problem (NCP) for cases with and without physical separation.

Link	Mode	1	2	3	4	5
Without physical separation	$v_{a}$	43.69	109.46	81.006	72.156	153.15
	${\bar{v}}_{a}$	256.31	90.54	0	346.85	346.85
With physical separation	$v_{a}$	90.18	88.55	55.36	123.37	178.73
	${\bar{v}}_{a}$	209.82	111.45	71.03	250.24	321.27

**Table 3 ijerph-17-03704-t003:** Route flows of the NCP for cases with and without physical separation.

Case	OD	Route	Route Flow of Vehicles	Route Flow of E-Bikes	Route Cost of Vehicles	Route Cost of E-Bikes
Without physical separation	1-5	1-3-5	12.70	0	**415.08**	429.25
		1-4-5	31.00	256.31	**415.08**	**415.08**
	2-5	2-3-5	68.30	0	**382.08**	396.24
		2-4-5	41.16	90.54	**382.08**	**382.08**
With physical separation	1-5	1-3-5	22.87	11.33	**184.61**	**184.61**
		1-4-5	67.31	198.48	**184.61**	**184.61**
	2-5	2-3-5	32.49	59.70	**155.81**	**155.81**
		2-4-5	56.05	51.76	**155.81**	**155.81**

Numbers in bold means the cost of each used route (including vehicles and E-bikes) is equal to its minimum travel cost.

**Table 4 ijerph-17-03704-t004:** Indicators of the NCP for cases with and without physical separation.

Index	Demand of Vehicles $\left(\sum_{w}\sum_{i\in R^{w}}\sigma_{i}^{w}f_{i}^{w}\right)$	Demand of E-Bikes $\left(\sum_{w}\sum_{i\in R^{w}}\sigma_{i}^{w}{\bar{f}}_{i}^{w}\right)$	System Total Travel Time $\left(\sum_{a\in A}(v_{a}T_{a}+{\bar{v}}_{a}{\bar{T}}_{a})\right)$
Without physical separation	153.15	346.85	148,760.5
With physical separation	178.73	321.27	63,544.0

**Table 5 ijerph-17-03704-t005:** Sensitivity analysis regarding parameters related to interactions between vehicles and E-bikes.

$\gamma_{m}$	$\gamma_{e}$	Total Travel Time of Vehicles $\left(\sum_{a\in A}v_{a}T_{a}\right)$	Total Travel Time of E-Bikes $\left(\sum_{a\in A}{\bar{v}}_{a}{\bar{T}}_{a}\right)$	System Total Travel Time $\left(\sum_{a\in A}(v_{a}T_{a}+{\bar{v}}_{a}{\bar{T}}_{a})\right)$
0.05	0.5	28,934.3	41,812.6	70,746.9
0.1	1	34,368.3	51,494.2	85,862.6
0.15	1.5	40,223.1	62,836.3	103,059.4
0.2	2	45,795.2	75,477.3	121,272.5
0.25	2.5	49,974.9	88,623.3	138,598.3
0.3	3	49,546.9	99,213.6	148,760.5

**Table 6 ijerph-17-03704-t006:** Link parameters of the Nguyen–Dupuis network.

Link	$T_{a}^{0}$	$C_{a}$	${\bar{T}}_{a}^{0}$	${\bar{C}}_{a}$	Link	$T_{a}^{0}$	$C_{a}$	${\bar{T}}_{a}^{0}$	${\bar{C}}_{a}$	Link	$T_{a}^{0}$	$C_{a}$	${\bar{T}}_{a}^{0}$	${\bar{C}}_{a}$
1	7	70	14	140	14	5	70	10	140	27	4	28	8	56
2	9	56	18	112	15	4	28	8	56	28	4	28	8	56
3	9	70	18	140	16	6	14	12	28	29	4	70	8	140
4	9	28	18	56	17	5	70	10	140	30	9	28	18	56
5	9	56	18	112	18	9	70	18	140	31	8	56	16	112
6	11	56	22	112	19	4	70	8	140	32	4	70	8	140
7	12	56	24	112	20	9	70	18	140	33	4	70	8	140
8	5	37	10	74	21	9	70	18	140	34	9	56	18	112
9	7	70	14	140	22	14	56	28	112	35	7	14	14	28
10	12	56	24	112	23	5	37	10	74	36	14	56	28	112
11	12	42	24	84	24	9	42	18	84	37	11	56	22	112
12	9	42	18	84	25	5	28	10	56	38	9	28	18	56
13	12	42	24	84	26	9	28	18	56					

**Table 7 ijerph-17-03704-t007:** Origin–Destination (OD) pairs and the corresponding demand.

OD No.	O	D	Demand	OD No.	O	D	Demand
1	1	2	210.00	10	4	2	320.00
2	1	3	430.00	11	4	3	110.00
3	1	8	320.00	12	4	8	210.00
4	2	1	210.00	13	8	1	320.00
5	2	4	320.00	14	8	4	210.00
6	2	12	50.00	15	8	12	60.00
7	3	1	430.00	16	12	2	50.00
8	3	4	110.00	17	12	3	40.00
9	3	12	40.00	18	12	8	60.00

**Table 8 ijerph-17-03704-t008:** Link flows of the NCP for the Nguyen–Dupuis network.

Link	$v_{a}$	${\bar{v}}_{a}$	Link	$v_{a}$	${\bar{v}}_{a}$	Link	$v_{a}$	${\bar{v}}_{a}$
1	83.84	335.47	14	75.87	332.90	27	26.34	60.69
2	66.79	473.90	15	24.98	74.76	28	34.38	390.82
3	109.29	175.43	16	25.79	112.23	29	61.04	434.67
4	15.97	279.31	17	77.65	332.82	30	16.56	277.95
5	51.53	249.78	18	74.01	239.21	31	46.34	250.42
6	47.25	231.44	19	3.22	226.62	32	8.14	210.53
7	56.63	115.54	20	111.05	174.44	33	60.72	451.50
8	17.06	450.78	21	70.88	255.23	34	67.33	469.30
9	86.63	336.73	22	70.93	477.68	35	26.24	102.18
10	56.37	107.38	23	16.83	459.42	36	68.68	493.60
11	74.61	305.48	24	64.80	162.84	37	46.86	236.39
12	65.86	145.52	25	36.06	359.91	38	47.25	231.44
13	78.20	281.28	26	46.86	236.39			

**Table 9 ijerph-17-03704-t009:** Route flows of the NCP for the Nguyen–Dupuis network.

OD	OD Demand	Route	Route Flow of Vehicles	Route Flow of E-Bikes	Route Cost of Vehicles	Route Cost of E-Bikes
1-3	430.00	1-11-14-19-31	2.36	129.10	246.89	246.89
		1-11-15-29-31	0	0	246.89	246.89
		1-12-25-29-31	19.00	19.13	246.89	246.89
		1-12-26-37	46.86	126.39	246.89	246.89
		2-35-14-19-31	0	12.39	246.89	246.89
		2-35-15-29-31	0	74.76	246.89	246.89
4-2	320	7-11-14-18-20	25.49	0	269.85	281.06
		8-25-29-30	16.56	277.95	269.85	269.85
		8-25-29-32-18-20	0	0	269.85	281.06

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
