# Peer review of "A Combined Modal and Route Choice Behavioral Complementarity Equilibrium Model with Users of Vehicles and Electric Bicycles"

_ijerph, 2020, doi:10.3390/ijerph17103704_

Round 1

Reviewer 1 Report

The authors proposed a model to capture the mode choice and route choice behavior in the context of electric bicycles.  They found an increase in electric bicycle choice when the network is high congestion. They also found a significant association between travel time and the separated lanes of vehicles. The study’s hypothesis and concept are interesting, but I am afraid that the manuscript still has several issues that need to revise.  Here I would like to offer some comments as follows

  1. The authors should provide much more discussion on why the study was necessary. I accept that the proposed model and its hypothesis are good, but I feel that the application of this model on traveler behavior has not been spelled out convincingly.
  2. The authors mentioned that they developed the BPR function, but there are no related introduction and necessary references.
  3. The numerical example was based on the assumption flow in the traffic network. However, it is not clear what type of vehicle is used to analyze with E-bike in the network. Do the authors mean “vehicle” or “car” throughout the manuscript?
  4. The model formulations should re-write briefly. For example, the proofs may be described in the Appendix.
  5. There are several linguistic errors, presumably due to the fact that none of the authors is a native English speaker.  It is recommended that the authors seek editorial assistance from a native English colleague to check the manuscript.

Author Response

We are very grateful to your valuable comments on our paper. We have studied the comments carefully and revised the paper accordingly. Please see the attachment to check the detailed responses to the comments.

Reviewer 2 Report

The manuscript proposes a traffic assignment model by incorporating e-bike travel. The idea is straightforward, but the methodology was somewhat new to me, which is formulated as a Complementary Problem. The authors claim that the proposed approach can solve both mode choice and route choice simultaneously, but I have to partially agree because one reason is that the mode choice process in transportation modeling typically involves statistical modeling such as discrete choice model by relating to individual traveler’s attribute and availability (car ownership, value of travel time, price, convenience, and others) of the alternative modes. In this approach, the authors add the e-bike choice into the traffic assignment procedure by adding a new BPR, implicitly assuming all vehicle travelers can be potential e-bike users. Some people may argue with this point because people who do not own cars need to use e-bike.

I have noticed that the assignment algorithm starts with a preset of routes for vehicle and e-bike, which is not a problem to me, but I was wondering if it is one of characteristics of using Complementary Problem. For example, other traffic assignment algorithms update feasible OD paths at every iteration. I like the idea to come up with the out-of-pocket costs to incorporate other factors in to BPR, but I would address the challenge to calibrate such factors into BPR parameters. It would be good if the authors mention how to calibrate the e-bikes’ link capacity, travel time curve, and out-of-pocket cost because I can imagine difficulty. Overall, I think it is worth investigating. Thank you!

Author Response

(The authors gave the same response as above.)

Reviewer 3 Report

Dear editor and authors,

This paper has addressed the multimode and traffic assignment problem of heterogeneous travel behavior by using a complementarity problem. Moreover, modified BPR functions have been created to capture interactions between the modes of e-bikes and motorized vehicles and the effects of the characteristics of each mode on link travel time. In addition to travel time, out-of-pocket costs related to security, environment, distance, fuel or electricity price etc. affecting users’ mode choice behavior have been taken into consideration in their model. The authors have also developed a newton method for solving the model.

The English language is good, only some very small mistakes could be taken care of. 

Author Response

(The authors gave the same response as above.)

Reviewer 4 Report

This paper has presented a new model characterising the way in which travellers tend to choose between motorised transport and E-Bikes. Although the topic is appealing, this research should undergo a considerable change before it is published.

Below are my comments on how the paper must be addressed.

  1. The paper’s English needs to be revised by a professional proof-reader.

  1. The authors should add a section called “Literature Review” after the Introduction to discuss the current works there. The present paper lacks a proper and comprehensive review of the literature.

  1. It should be discussed in the Literature Review section whether the current studies have implemented similar mode choice models, and how the model presented in this study differs to theirs.

  1. The mode choice model offered in this paper is over-simplified. The way in which people decide which mode to use is far complex than what this study suggests.

For instance, this study assumes that as the number of vehicles on roads grows, people switch to use E-Bikes straightaway (Figure2). It is not what happens in reality. Since many years ago, urban congestion has been ever increasing across the world. However, the rate in which people have shifted to active transport systems or systems such as E-Bikes has never been similar to what is suggested in this article and Figure2.

Many trips are not suited to active transport or even E-Bikes. Most people need to carry heavy goods or do not feel comfortable to use E-Bikes rather than their private cars. All these matters make private vehicles very attractive so that it is not easy to convince travellers to use other transport modes.

Although this paper attempts to take into account factors such as comfort, security, weather or etc as out of pocket costs, the results of the model presented in Figure 2 and Figure 3 show the model fails to capture the impact of these parameters adequately. 

Figure 2 and Figure 3 are portraying EBikes as extremely popular modes of transport where people shift to use them rather than vehicles as soon as demand increases. For instance, Figure2 suggests when number of cars are increased from 200 to 300 (50%), EBike use grows almost 30%. It is not realistic and has never been observed any where in the world.

People’s mode choice preferences are far more complicated to be captured by such simplistic assumptions and models. The authors should thoroughly investigate the current mode choice models in the literature and come up with a more realistic model drawing on the findings of previous studies.

  1. Page5, Line146, What does NCP stand for?
  2. Page5, Line 173, How did you come up with equations 12 and 13. It should be clearly stated.
  3. In page12, what Figure3 and each of the graphs are trying to convey should be comprehensively discussed in the text.

Author Response

(The authors gave the same response as above.)

Round 2

Reviewer 4 Report

Authors have satisfactorilly addressed the comments and improved the overall quality of the article.

I hope in their future work we will see a more rigorous and solid research.